# The Role of Social and Cultural Values in Pandemic Control in a Chinese Community: An Ethnographic Study on the Construction and Stigmatization of “Others” in Severe Acute Respiratory Syndrome (SARS) and COVID-19 in Hong Kong

**DOI:** 10.3390/ijerph192013517

**Published:** 2022-10-19

**Authors:** Judy Yuen-man Siu

**Affiliations:** 1Department of Applied Social Sciences, Faculty of Health and Social Sciences, The Hong Kong Polytechnic University, Hong Kong, China; judy.ym.siu@polyu.edu.hk; 2Interdisciplinary Centre for Qualitative Research, The Hong Kong Polytechnic University, Hong Kong, China; 3Research Centre for Sharp Vision, The Hong Kong Polytechnic University, Hong Kong, China

**Keywords:** social and cultural values, others, social control, epidemic control, Severe Acute Respiratory Syndrome (SARS), COVID-19, Hong Kong

## Abstract

Background: Studies have widely reported that social and cultural values serve as constraints in controlling the spread of an epidemic. However, I argue that a social and cultural value system is a double-edged sword and can motivate people’s preventive health behaviors. Few studies have examined the positive role of social and cultural values in promoting epidemic control. Methods: Using the severe acute respiratory syndrome outbreak in 2003 and the COVID-19 pandemic that began in 2020 in Hong Kong as examples, the present study performed participant observation in Hong Kong from January to June 2003 and from January 2020 to May 2022; in-depth individual semi-structured interviews were conducted with 70 participants between February 2021 and March 2022. Results: Social and cultural values serve as informal social control mechanisms in manipulating people’s adoption of preventive health behaviors that can assist in epidemic control. Specifically, the construction and stigmatization of the “others” groups and the traditional cultural values based on the capitalist ideology were noted to facilitate control measures against the two outbreaks in Hong Kong. Conclusion: These two outbreaks reinforced the embedded social and cultural values of the capitalist ideology of Hong Kong, which increased the vulnerability of disadvantaged social groups to stigmatization.

## 1. Introduction

The Severe Acute Respiratory Syndrome (SARS) outbreak in 2003 and the COVID-19 pandemic that began in 2020 are two major public health emergencies, as noted by the World Health Organization [1,2,3,4]. Both infectious diseases are caused by the same virus SARS-CoV [5,6]. The SARS outbreak infected a total of 8098 people globally and 774 died [5]. SARS was first reported in Asia in February 2003 and was then spread to North America, South America, and Europe [5]. Hong Kong is one of the cities that suffered from the most severe attack of SARS, with a travel advisory issued to Hong Kong by the WHO from 2 April 2003 to 23 June 2003 [7]. The SARS outbreak was fully contained in July 2003 in Hong Kong [7] and the outbreak ended globally in the same year. The SARS outbreak in Hong Kong ended with 1755 people infected, 299 of which died [8]. The COVID-19 pandemic, however, has been even more tragic worldwide since 2020. As of 7 July 2022, there have been about 550,218,992 confirmed cases of COVID-19, including 6,343,783 deaths globally [9]. In Hong Kong, there were 1,262,122 cases of infection, of which 9407 were terminal on the same date [10].

The social environment, which involves the social processes as well as the interrelationship between individuals and groups in society, was suggested as a significant factor in affecting the outcomes of epidemics [11]. Studies have indicated that the social and cultural values of a society have a major influence on the control of epidemics [12,13,14]. Past studies show that stigmatization of those who have been infected has led to public panic, but has achieved little to control the spread of epidemics of HIV/AIDS, tuberculosis, and cholera [12,13,14]. In addition, those who failed to fulfill the social norms, such as the poor, were shown to be more vulnerable to stigmatization in these epidemics [13,14,15]. Socially disadvantaged groups are often stigmatized scapegoats in such epidemics as well [13,15]. The stigmatized vulnerable groups are often the “others” of society, who are opposite to the mainstream of society—“us”—according to Goffman (1968). “Others” are the social deviants who fail to fulfill the social and cultural norms, therefore they were often secluded from the mainstream of society [16]. Stigmatization, as shown by these studies, is noted as an important factor contributing to the spread of epidemics because the “others” groups would conceal themselves on one hand and would resist because of their extensive seclusion experienced from the social mainstream on the other [12,13,14,16].

Studies have thus widely reported that social and cultural values can serve as constraints in controlling the spread of an epidemic [12,13,14,17,18]. Similar to these past studies, new epidemics such as SARS and COVID-19 have had remarkable social and cultural implications in Hong Kong and have demonstrated Hong Kong’s embedded social and cultural contours. However, as argued in this article, social and cultural values related to capitalism have played an opposite but major role in controlling the spread of SARS and COVID-19 in Hong Kong. The social and cultural values of Hong Kong were manifested and reinforced during these outbreaks and, by manipulating people’s behavior, served as important social and cultural agents in controlling the spread of these two epidemics. As I argue in this article, the embedded social and cultural values of Hong Kong, the social construction and stigmatization of different groups of “others,” and the construction of new social norms entailing the use of face masks in public areas have all played a favorable role in the control of the SARS and COVID-19 outbreaks in Hong Kong.

### Significance

It was suggested that there is an imbalance of epidemiological research with a strong focus on a physical environment approach, whereas research on the social environment in affecting epidemic outcomes is not well established [11]. Social and cultural value systems were popularly addressed as impeding factors for successful epidemic control [12,13,14,17,18]. However, I would argue that a social and cultural value system is a double-edged sword that can motivate people’s preventive health behaviors and few studies have examined the positive role of social and cultural values in promoting epidemic control. Furthermore, epidemiological studies applying the concept of “others” are scant; how stigma in epidemics is related to embedded social and cultural values of a society remains unknown. To fill this research gap, this study applied an ethnographic approach to conduct an investigation in Hong Kong during the 2003 SARS epidemic and the current COVID-19 pandemic to determine how the embedded social and cultural values of an area and how the construction of “others” can serve as a social control mechanism to motivate people’s preventive health behaviors and thus facilitate epidemic control.

## 2. Methods

### 2.1. Design and Study Area

An ethnographic method involving participant observation and in-depth individual semi-structured interviews was adopted in this study. This study was conducted in Hong Kong during the SARS outbreak from January to June 2003 and during the COVID-19 pandemic from January 2020 to the end of the fifth wave of infections in May 2022.

### 2.2. My Reflexivity in This Research

I was born and raised in Hong Kong and I was there when the SARS outbreak and the COVID-19 pandemic hit Hong Kong. During the SARS outbreak, I was finishing my Master’s degree thesis in Hong Kong, which is about the use of *qigong* and *tai chi* among chronically ill patients in Hong Kong. At the same time, I was amazed by how the SARS epidemic had manifested in the embedded social and cultural landscapes of Hong Kong. Therefore, I was also preparing the proposal for my doctoral program research, which aims at researching the social and cultural landscapes of the SARS epidemic in Hong Kong. The ethnographic data concerning the SARS outbreak in this article are mostly from my doctoral thesis [19].

SARS was a special experience for me, not only because it was the first and most severe epidemic that I had ever encountered, but also because it dramatically changed my understanding of how an epidemic can interact with a society’s social and cultural norms and impact social life. SARS is the first epidemic that made me anxious about getting infected, not only because it was portrayed as deadly, but getting infected would also have the social consequence of being discriminated against. I met many chronically ill patients when I was collecting data for my Master’s thesis and I was amazed by how anxious they were about SARS. Given their chronically ill status, they were even more nervous about being discriminated against, after the news reported that a chronic renal patient was a “hidden virus spreader” and had spread the virus, which led to a community outbreak through the sewage and drainage system in a housing area. I also experienced some form of informal social control during the SARS outbreak, compelling me to use a face mask when I was in public areas, even though there was no law enforcement on face mask use. After the face mask advocacy by medical professors, I, like other people in Hong Kong, started to learn about the importance of using a face mask in public areas. People were no longer afraid of those who used a face mask, but instead, more people began using a face mask, making it a positive symbol to the people of Hong Kong. Although SARS was combated 6 months after the first outbreak in Hong Kong, it attracted my attention and interest as an anthropology student. This was when I started to become aware that, even though an epidemic is a medical phenomenon, it still cannot escape the social and cultural landscape that can affect people’s responses.

The COVID-19 pandemic that started in Hong Kong in early 2020 reminded me of the SARS outbreak. All of the preventive measures that I followed during the COVID-19 pandemic were primarily based on what I had learned during the SARS outbreak, especially using a face mask in all public areas. However, the COVID-19 pandemic has made these preventive measures even more stringent. For example, using a face mask during the SARS outbreak was mostly by personal will and an informal social control mechanism, such as peer pressure, but this practice became enforced by law during the COVID-19 pandemic. People became pickier about the quality of face masks during the COVID-19 pandemic. Hand hygiene using hand sanitizers was also put into focus during the COVID-19 pandemic as well. SARS and COVID-19 have many similarities in terms of the social and cultural aftermath. Similar to the SARS outbreak, the social phenomena during the COVID-19 pandemic mirrored the embedded social and cultural landscapes of Hong Kong. Stigmatization and discrimination still targeted socially disadvantaged groups, which implies that these groups still suffered from inequality, both socially and in terms of health, even twenty years after the SARS outbreak. This study was conducted in such a social context.

### 2.3. Data Collection

Participant observation was conducted in Hong Kong during the SARS outbreak from January to June 2003 and during the COVID-19 pandemic from January 2020 to the end of the fifth wave of infections in May 2022. In the participant observation method, a researcher typically immerses themselves in a culture and community by working and living with the research participants for a certain period [20]. Through this process, the researcher can observe, experience, and understand how beliefs are embedded in local cultures from an “insider’s” point of view [20], thus enabling them to achieve an “emic” understanding of the participants and to maximize the insider relationship to reduce the participants’ reactivity and sensitivity that can engender participant behavioral changes during fieldwork [20]. Participant observation also enables the collection of data that cannot be obtained from interviews [20]. In the present study, I maintained two ethnographic diaries to record observational data and feelings as a participant-observer during the two outbreaks. An ethnographic diary of the SARS outbreak was maintained to record my observational data and experiences during the SARS outbreak from January to July 2003; this diary was retrieved when conducting this study. Another ethnographic diary of the COVID-19 pandemic has been maintained since January 2020; although this ethnographic diary is still ongoing, this article adopts the ethnographic data until the end of the fifth wave of infections in May 2022. 

After the execution of participant observation, in-depth individual semi-structured interviews were conducted. Seventy participants were recruited and interviewed from February 2021 to March 2022 during the first to the fifth waves of the COVID-19 outbreak in Hong Kong. To investigate the effect of social and cultural values on people’s health behaviors during the two outbreaks, participants were sampled through purposive sampling on the basis of the following criteria: (1) having lived in Hong Kong since the SARS outbreak in 2003; (2) having experienced both the SARS and COVID-19 outbreaks in Hong Kong; (3) being aged 18 years or older in 2003; (4) having been born and educated in Hong Kong; (5) being able to satisfactorily communicate in Cantonese; and (6) being Chinese in ethnicity. These sampling criteria ensured that I recruited participants with first-hand experience of the two outbreaks in Hong Kong. As the ethnographic data of the SARS outbreak was collected in 2003, recruiting participants who have first-hand experience of the SARS outbreak in Hong Kong could complement the data on the SARS outbreak. Their first-hand experience of the COVID-19 pandemic in Hong Kong could also allow them to compare their experiences of the two outbreaks. Sampling those who are born and educated in Hong Kong and are Chinese in ethnicity ensured that the participants have extensive social and cultural exposure in Hong Kong, which is important for this study, which investigates how the social and cultural values of Hong Kong served to control the SARS and COVID-19 from spreading. Some interviews and ethnographic data related to the SARS outbreak were obtained from my doctoral thesis [19] and these data complemented the data collected in this study.

The participants, 29 men and 41 women, were aged 36–78 years at the time of their interviews and they were recruited from different locations. Twenty participants were recruited from a university in Hong Kong during the first stage of the recruitment process. However, recent studies in Hong Kong note that those who are living in the wealthiest areas were 65% less likely to be cases in sizable infection clusters than those living in the poorest areas, and those living in private housing were less likely to become infected at work than those living in public, subsidized housing [21]. Significant adverse effects of multimorbidity on severe COVID-19 were more notable among cases living in poor districts than those living in wealthier districts [22]. People with lower educational attainment and living in public, subsidized housing were also noted to require longer periods of time for the diagnosis of COVID-19 [23]. To ensure a wide demographic and sociocultural mix among the participants, an additional 50 participants were recruited from the community, including the labor unions and leisure classes of the Hong Kong Federation of Trade Unions, covering a wide spectrum of occupations [24]. As this field site is one of the largest labor organizations in Hong Kong, recruiting participants from it ensured that they were representative groups of the working population. The demographic background of the 70 participants is shown in Appendix A.

An interview question guide (see Appendix B) was developed prior to the interviews to ensure that the interview discussion focused on the research questions. The interview questions were aimed at investigating the experiences of the participants during the SARS and COVID-19 outbreaks, with a particular focus on determining how the participants’ perceptions, behaviors, and experiences during the two outbreaks were related to the social and cultural values of Hong Kong. The questions were open-ended to ensure that the participants had a high degree of flexibility in expressing their views, feelings, and experiences [20]. I conducted all the interviews to ensure interview consistency. The interviews were conducted in Cantonese, which is the mother tongue of the participants and myself, to facilitate an active interaction process. Because of the COVID-19 pandemic, Skype for Business and Zoom were used to conduct the video interviews. The interviews were audio-recorded with the consent of the participants.

### 2.4. Ethics

Ethical approval was obtained from the Institutional Review Board of the Hong Kong Polytechnic University prior to this study (reference number: HSEARS20200924003). All participants were adequately informed about the purpose of this study, and written informed consent was obtained from each participant prior to the interviews. All interviews were conducted anonymously and each of the participants was represented by a code in the data to ensure confidentiality. All collected data were stored in password-protected computer files that could only be accessed by the researcher.

### 2.5. Data Analysis

Rapid data analysis was conducted during the interviews to determine aspects that required further investigation [25]. The interviews were transcribed verbatim, and the raw text of the interviews was read thoroughly to achieve familiarity with their content. The raw text was then reread to discover possible themes [26]. The interview transcriptions were analyzed line by line through an inductive coding process, which enabled the identification of the participants’ thinking and behavioral patterns [27]. Furthermore, the interview transcriptions were segmented into smaller meaning units [26]. The segments were labeled and then classified into codes in the coding table [26]. The codes were organized and grouped under categories. Upper-level categories were identified on the basis of the research questions, and in vivo coding was conducted [26]; recurrent categories were highlighted. Overlapping codes and categories were grouped after repeated examination and comparison to form larger themes [26]. The codes, categories, and themes derived from the collected data as well as the supporting interview quotes were documented in a coding table [25], and data saturation was achieved. A validity check of member checking was performed with the participants, and they were invited to read their transcribed interviews through email to ensure that they agreed with the transcribed interviews and to achieve an emic understanding [25].

The interview quotes used in this paper were translated into English. Back-translation to Chinese was conducted to ensure that the original meanings were not distorted.

During the participant observation process, observational data and my experiences were recorded in an ethnographic diary for cross-analysis. The ethnographic data were transformed into codes based on an inductive process and were then classified into categories and themes in another coding table. Memos were used to record ideas and commentary during the coding process. The analytical procedures, codes, and findings were documented in a codebook to ensure the consistency and accuracy of the collected data.

## 3. Results

Throughout the 70 interviews, the literal use of “COVID-19” was rare among the participants. All participants referred to “COVID-19” as “pneumonia,” as the term “coronavirus pneumonia” was popularly used in Chinese mass media and government press conferences instead of “COVID-19.”

Two social and cultural agents—(1) the social construction and stigmatization of “others” and (2) traditional cultural values—were identified as informal social control mechanisms for manipulating people’s adoption of health behaviors favorable to the control of the SARS and COVID-19 outbreaks. A concept map of these agents is outlined in Appendix C. These agents are described in detail as follows.

### 3.1. Social Construction and Stigmatization of “Others”

“Others” is a social construction according to the social and cultural norms of society and those who fail to fulfill social and cultural norms are regarded as “others” (Goffman, 1968). The identification of “others” was common during the SARS and COVID-19 outbreaks in Hong Kong. During the two outbreaks, five types of “others” were constructed: “diseased others,” “deviant others,” “geographical others,” “occupational others,” and “ethnic others,” and they were stigmatized and scapegoated as “dangerous” for having a higher risk of infection. The construction of these types of “others” engendered an informal social control mechanism for manipulating people’s health behaviors suitable for outbreak control.

#### 3.1.1. Social Construction of “Diseased Others”

##### Infected Individuals

Individuals who were infected with SARS or COVID-19 constituted one group of “diseased others” because they were the social deviants who became sick. These individuals experienced stigmatization and discrimination because SARS and COVID-19 were portrayed as “highly contagious” and “deadly”:

SARS and pneumonia [COVID-19] are highly infectious and deadly, so I avoid those who are infected. These two diseases are worse than AIDS because AIDS is not that contagious. Yuen Kwok-yung [Professor in microbiology at the Medical School of the University of Hong Kong] said that pneumonia can be transmitted by air, and even the WHO does not deny this. Some news reports have also reported that you can be infected by simply walking past virus carriers for a few seconds only. Therefore, if I know someone is infected, I would definitely avoid them.(P2)

The participants perceived that contracting SARS or COVID-19 infection was associated with the infected person’s negative personal attributes and blamed the infected person for contracting the virus. This association and blame constituted a remarkable form of informal social control for people’s health behaviors:

One of my neighbors contracted SARS during [the] SARS [outbreak]. I remember when I learned that my neighbor had [contracted] SARS. My immediate response was to feel horrified and afraid, and I was concerned about whether I had already caught the virus from them. My other neighbors and I scolded the infected neighbor loudly outside their flat, blaming them for bringing this disease so close to our homes. I believe that, if you can maintain good hygiene, you will not be infected [with SARS]. Those who were infected with SARS are likely dirty and have low quality… Now, with regard to pneumonia [COVID-19], I feel even more nervous because if any of the neighbors are infected, we would all be sent to quarantine camps. Therefore, I have been maintaining an eye out for neighbors who appear to have a higher possibility of being infected and then avoid them. Those who are poor, are dirty, and have low quality are likely to be at a higher risk of being infected.(P26)

During the COVID-19 pandemic, the blame placed on infected individuals was even more severe than that observed during the SARS outbreak because of the Hong Kong government’s quarantine policy:

Three neighbors in my block were infected with pneumonia [COVID-19] in December [2020]. You know, if individuals from two flats in the same block are infected, then all the people living in the block would be forced to undergo compulsory testing. Because of these three neighbors, all of us were forced to undergo testing. I did not mind undergoing testing, but that affected my job because my boss immediately asked me not to come back to the office. I have no money if I cannot work because my salary is calculated on the basis of working days. These three neighbors have caused me to lose my salary. They are really ‘bad luck gods’ [a Cantonese slang that denotes someone who brings bad fortune to others].(P19)

The pandemic control measures implemented by government health institutions constituted a social control mechanism in motivating people’s health behaviors and made infection a collective responsibility. The Hong Kong government tightened its pandemic control policy during the fourth wave of the COVID-19 outbreak by requiring that, if two different units have confirmed cases of infection, all residents living in the same housing block as these two units must undergo COVID-19 testing under the Prevention and Control of Disease (Compulsory Testing for Certain Persons) Regulation. Buildings with confirmed cases were listed in the public domain. In January 2021, certain districts with residents from lower socioeconomic classes, such as Yau Tsim Mong District and Sham Shui Po District, were subjected to a highly strict pandemic control policy in view of the local outbreak in these districts. All of the residents of these districts were required to undergo compulsory virus testing. Thus, avoiding COVID-19 infection was no longer an individual concern but rather a collective responsibility that even involved the uninfected. Consequently, infected individuals were blamed in many cases. Such scapegoating and collective responsibility served as a type of informal social control mechanism for shaping people’s health behaviors to conform to the pandemic control policy:

It is very important to prevent myself from being infected. If [I] get infected, not only will my family members be sent to an isolation camp (quarantine camp), but my neighbors will also be required to undergo compulsory testing. If the neighbors know that I am infected, then I will be blamed. Similarly, if I know that a neighbor is infected, I will blame them as well because I will be forced to undergo compulsory testing. To avoid being blamed, I must be more cautious for evading infection.(P4)

Because SARS and COVID-19 have collective characteristics, several participants indicated that they would feel guilty if they had infected others, which motivated them to adhere to pandemic control measures:

I am not quite afraid of being infected, but I am considerably afraid of infecting others, especially because I can never know if I am a hidden virus carrier. If I infect others, I will make them and those who had contact with them suffer isolation [quarantine]. If it is just me becoming sick, that’s fine because it is my own business. However, being infected is a collective matter and has a chain effect; involving others is what I am most afraid of. Therefore, I am very cautious to avoid being infected, not just for my own health but, more importantly, for not involving others.(P58)

As a pandemic control measure, government health institutions publicized the residential information of infected individuals. This measure empowered the public to exercise informal social control over individuals:

The building where I live had three [COVID-19] cases. When my colleagues learned about this from the Internet, they immediately reported it to my boss, and I was then asked to work from home immediately. Even my friends avoided me. I understood that everyone would be afraid of me, but I still felt bad. If I was the one to be infected, I would find it easier to understand [the situation]; however, it was my neighbors who were infected, not me. I believe that I did not deserve to be shunned.(P13)

Some residence management offices posted the residential information of infected individuals on their public facilities such as bulletin boards during the SARS and COVID-19 outbreaks, which made it easy to identify these individuals and resulted in their stigmatization and discrimination. In the two areas that I lived in during the two outbreaks, publicizing the residential information of infected individuals was a popular practice. This information often constituted the basis for an informal appraisal of the “performance” and “qualities” of the residents. Residents attempted to establish a correlation between the morbidity rate and the demographic characteristics of other residents (mainly in terms of socioeconomic factors such as occupation and social hierarchy):

Neighbor A:“See? Block 45 has a new infection again! The residents living in that block must have low quality.”

Neighbor B:“I don’t feel surprised though. The flat size in that block is small, so the people living there are likely to be not as rich as us. If they are not rich, then their jobs may not be good, and they may not be of good quality; otherwise, they would be able to get a good job and high salary and be able to afford to live in bigger units like us. If they were of high quality, then they would be cleaner. Do you recognize that all infections are happening in blocks with a small flat size?” [SARS_G2]

The practice of posting the residential information of infected individuals resulted in their stigmatization as “diseased others.” The stigmatization of infected individuals was an informal social control mechanism for manipulating people’s health behaviors during the outbreaks. Even residents who were not infected but were living in the same building as infected individuals could experience potential stigmatization and discrimination because of the collective responsibility associated with infection. Such stigmatization originated from Hong Kong’s capitalist ideology. Individuals from a lower social hierarchy and of lower socioeconomic status were more vulnerable to stigmatization. Consequently, a correlation between infection and negative social attributes was constructed, which reinforced the existing cultural values and capitalist ideology of Hong Kong.

##### Those Who Had Respiratory Symptoms

People with respiratory symptoms but not infected with SARS and COVID-19 were secluded in particular during the COVID-19 outbreak. A number of participants shared that their respiratory symptoms had made visiting their doctor difficult during the outbreaks. Such experience could serve as a social control to motivate people to stay healthy:

It is a bad idea to get sick during COVID [COVID-19], because no doctor will want to see you. I had a fever and diarrhea two months ago, and it was a nightmare for me because no doctors were willing to see me. Some would ask you to have a proof of a virus-negative result from a deep throat saliva test before they see you, but taking a deep throat test requires [several] days to get the result, and I would have died before I could get that. Others would just ask you to go to the emergency room. Indeed, many diseases can give you a fever and diarrhea, but in COVID times, all people, including doctors, would just think that you are getting COVID. It is a sad thing, because you can never imagine that you are being treated in such a way by doctors, who are expected to help those who are sick. The whole experience in COVID is much different from SARS. It is especially important to stay healthy in these COVID times.(P69)

##### Chronically Ill Patients

Another group of “diseased others” comprised chronically ill patients, who were stigmatized as a high-risk group, according to the participants of this study:

News reports have stated that chronically ill patients can carry a high viral load, become “super-spreaders,” and infect many people because they are weakened [more than others are] when infected [with COVID-19]. I have avoided those whom I know to have chronic conditions since the [COVID-19] outbreak. Indeed, I learned about this during [the] SARS [outbreak]. I remember that a chronically ill patient was a super-spreader at that time, and he had spread the virus through his excreta when he was living in Amoy Gardens [a private housing area]. He caused many people living there to become infected.(P10)

The relationship among chronically ill patients, infectious diseases, and super-spreaders has been socially constructed since the SARS outbreak. Because a patient with the chronic renal disease was the “index case” for the community outbreak of SARS in Amoy Gardens, patients with this disease became scapegoats who were blamed and stigmatized as “super-spreaders” of SARS. This stereotype has become embedded in the perceptions of people in Hong Kong and has reappeared during the COVID-19 pandemic. The stigma of being super-spreaders can damage the social network of chronically ill patients during the COVID-19 pandemic, thereby further marginalizing them as “others”:

I believe that chronically ill patients are dangerous because they are super-spreaders. They can carry a high viral load and can infect many people. Because pneumonia [COVID-19] is very similar to SARS, I believe that chronically ill patients might become super-spreaders again. It is better for me to stay away from them. Although they may not be super-spreaders at this time, they still have to go to hospitals and clinics often. You know, hospitals and clinics have numerous bacteria and viruses, so it is safer for me to stay away from them these days.(P17)

#### 3.1.2. Social Construction of “Deviant Others”

“Deviant others” comprised those who failed to fulfill social and cultural norms—both old and new—during the SARS and COVID-19 outbreaks. The social construction and stigmatization of “deviant others” during the two outbreaks served as an informal social control mechanism that motivated people’s preventive health behaviors. Such stigmatization followed and reinforced the social and cultural norms of Hong Kong.

##### Not Wearing a Face Mask

One of the major groups of “deviant others” comprised those who failed to wear a face mask in public areas. The wearing of face masks in public areas became a new social norm in Hong Kong during the SARS and COVID-19 outbreaks. Failure to conform to this social norm was perceived to be associated with negative social attributes:

If I see anyone not wearing a face mask, I walk away from them and give them a dirty look. I understand that no one likes wearing a face mask, but everyone should make efforts to fight against pneumonia [COVID-19] by wearing a face mask. Wearing a face mask is not only for your own health, but also for others’ health. All of us should have learned about this from [the] SARS [outbreak] already. At that time, no one would force you to use a face mask, but everyone would use a face mask in public areas, even though there was no face mask law at that time, because everyone understood that wearing a face mask was socially responsible. If someone does not wear a face mask, that means that they do not have [a sense of] civic responsibility and consideration for others. I would let these people know explicitly that they are so disgusting.(P24)

In addition to those who failed to wear a face mask in public areas, smokers were considered “deviant others” during the COVID-19 pandemic. The stigmatization of smokers during the COVID-19 pandemic was a continuation and reinforcement of the existing social norm that smoking is not socially preferable and was associated with the embedded negative stereotypes regarding smokers:

I walk away from those who are smoking on the streets because they have to take off their face masks when they smoke. They blow out air and saliva, so it is very dangerous to walk close to them. […] Smoking indeed is very annoying. Those who smoke are not good people, and they may be members of gangs, so it is impossible to ask them not to smoke unless you want to be beaten. The best way is to walk away from them.(P11)

The stigmatization of “deviant others” who failed to wear face masks in public areas was a remarkable social control mechanism under which the rights of such individuals were suspended during the COVID-19 pandemic. The service industry, including restaurants and retail stores, refused to provide services to those without face masks. Notices asking customers to wear a face mask before entering public facilities were easily noticeable. I witnessed bus drivers refusing to allow people without face masks from boarding their buses:

Bus driver:“You have to wear a face mask to get on this bus.”

Old lady:“I am sorry but I have no face mask now…”

Bus driver:“You cannot get on [the bus] without a face mask. If you have no face mask, you should stay at home. I cannot help.”

##### Not Following the Stay-at-Home Policy

The Hong Kong government encouraged a stay-at-home policy during the COVID-19 pandemic. In relation to this new policy, a new social norm of staying at home was constructed, constituting an informal social control mechanism for the participants:

Now, everyone is talking about staying at home on Facebook. The slogan of ‘I (health-care provider) stay at work for you (the public), you stay at home for us’ is really a form of social pressure. It sounds like, if you do not stay at home, you are bad. I dare not tell others that I still go out because I am working in a restaurant, and my job does not allow me to work from home. However, no one would care about the fact that many people cannot stay at home and work from home, but others would just think that you are harming society because you still go out.(P63)

The slogan of “I stay at work for you, you stay at home for us” during the COVID-19 pandemic served as a powerful social control mechanism for motivating the general public to remain confined to their homes as an infection prevention measure. With the construction of the new social norm of staying at home, those who failed to comply with the stay-at-home policy were stigmatized as lacking civic responsibility:

To help our society and to reduce the burden on doctors and nurses, I think we should stay at home as much as we can these days. Many doctors and nurses have been posting the slogan of ‘I stay at work for you, you stay at home for us’ on Facebook now. They have been asking for our help, so I think every responsible citizen should stay at home as much as possible. Those who are still going out and eating out are so irresponsible toward society! I don’t care if they are exposing themselves to harm and infection, but I care that they are harming others and increasing the workload of health-care providers by exposing themselves to infection.(P3)

To some participants, the aforementioned slogan promoted a sense of heroism, which motivated them further to conform to the stay-at-home appeal:

This is really a motivating slogan because this slogan not only praises doctors and nurses but also emphasizes to the public that everyone can contribute to the fight against the pandemic [COVID-19]. Every time I see this slogan, I feel that I am playing a very important role [in the fight] against pneumonia [COVID-19]. If I do not follow this slogan, then it would appear that I am not playing a role in this pandemic and not supporting doctors and nurses.(P42)

The aforementioned slogan enabled the construction of the new social norm of staying at home, which served as an informal social control mechanism for manipulating people’s health behaviors during the COVID-19 pandemic. Failure to conform to this social norm was perceived as deviancy. However, the stay-at-home appeal occasionally engendered difficulties for participants of low socioeconomic status. They found it difficult to fulfill the existing capitalist norms and new stay-at-home norm simultaneously, rendering them highly vulnerable to stigmatization during the COVID-19 pandemic:

I also want to stay at home. After all, who is not afraid of being infected? However, not all people can really stay at home. I still have to go out for work because I am living in a subdivided flat [a flat in which multiple families reside], so how can I afford to use a computer and the Internet to work from home? Also, not all jobs allow working from home. For example, many street cleaners, transportation workers, and service and catering workers have to go out to work. Not all people have the privilege [of being able to] work from home. However, working from home and staying at home have become new social values that you have to follow now because no one would think that you cannot stay at home.(P35)

##### Not Following the Takeout Policy

Eating takeout instead of dining-in was encouraged during the COVID-19 pandemic as a pandemic control measure and eating takeout became a new social norm that served as an informal social control mechanism:

Medical professors always encourage eating takeout [instead of dining-in]. Yuen Kwok-yung [Professor of microbiology at the Medical School of the University of Hong Kong] has said that dining in can be very dangerous because everyone takes off their face mask when eating. Now, so many people are buying takeout, and it [the encouragement for buying takeout] is just like a type of peer pressure; if you dine-in, you would appear strange. If someone dines in, I would think that they are very brave, but simultaneously, I would also think that they are not very responsible toward society.(P16)

Similar to the stay-at-home appeal, the encouragement of eating takeout presented difficulties for participants of low socioeconomic status:

It is very difficult for me to eat takeout because I work outdoors and do not have a stable workplace. Dining in is very important to me because this is the only moment that I can have some rest and can enjoy air conditioning. If I am forced to eat takeout, I can just go to parks to eat, but parks are very hot now. However, not wearing a face mask outdoors is against the law, so can I really eat in parks? Eating takeout looks very simple to many people, but it is very difficult for those who have to work outdoors, like me. It is a privilege if your job allows you to eat takeout.(P9)

##### People of Low Socioeconomic Status

Because of the appeals to stay at home and eat takeout, people of low socioeconomic status suffered from further stigmatization during both outbreaks. These “deviant others” belonged to socially disadvantaged groups and their socioeconomic status prevented them from adhering to the new social norms. The Hong Kong government’s pandemic control policies increased their vulnerability to stigmatization during the SARS and COVID-19 outbreaks. Thus, the stigmatization of the aforementioned social group during the two outbreaks was associated with their social attributes:

I believe that those who do not wear face masks belong to a low social class. They cannot afford to buy face masks, and they are not sufficiently educated to understand the importance of wearing face masks. Therefore, I believe that people from low social classes are a high-risk group. If you look at recent community outbreaks, most of them occurred in public housing estates [subsidized housing provided by the government for the low-income class] and in old buildings in poor districts. Have you ever seen the Mid-Levels and Kowloon Tong [residential areas for the high-income class] experience an outbreak? No, because the people living [there] are rich and highly educated, so they wear face masks and are more hygienic. Not only in the case of pneumonia [COVID-19], but also during the SARS outbreak, these infections never hit the rich areas.(P7)

The “deviant others” belonged to a low socioeconomic class and thus experienced double stigmatization. Their economic position prevented them from fulfilling the new social norms and existing social norms based on the ideology of capitalism in Hong Kong. The double stigmatization experienced by the participants from low socioeconomic classes not only demonstrated an informal social control mechanism for manipulating people’s health behaviors but also reinforced the embedded capitalist ideology. In the housing areas that I was living in during the SARS and COVID-19 outbreaks, gossip regarding neighbors who failed to wear face masks was widespread and such gossip always focused on the economic status of the neighbors:

Neighbor A:“That fat man’s family is really dirty. I cannot understand why they do not wear face masks. They just used tissues to cover their noses and mouths.”

Neighbor B:“I have heard that they are receiving CSSA (Comprehensive Social Security Assistance) and are waiting for public housing estates, so they may not be able to afford [buying] face masks.”

Neighbor C:“From their appearance and dressing, you can understand that they are poor. If they are not ‘eating government welfare’ [a Cantonese slang meaning receiving social welfare assistance from the government], how can they afford to live here? You know, the government pays the rent for them while they are waiting for public housing.”

Neighbor B:“We taxpayers keep subsidizing these lazy people, but they never make any contribution [to society]. They just keep taking resources from society, and now, they are harming society by not wearing face masks. If they become infected, all of us will be affected, and we will all be sent to an isolation camp [quarantine camp].” (COVID_G5)

##### Older People

Older people constituted another group of “deviant others” during the two outbreaks. The stigmatization of older people was related to their failure to follow the new social norm of wearing a face mask and existing social norms:

I believe that older people are the most uncooperative group. I can always see older people pull their face masks down to the chin. Indeed, those who spit on the streets are mostly older people. During [the] SARS [outbreak], older people were a high-risk group, and they were very “toxic” because they were hidden virus spreaders. I believe that this logic [also] applies to the pneumonia [COVID-19] outbreak because many cluster outbreaks are occurring among older people in nursing homes. I would walk away from them [older people], as you can never know if any of them is a hidden virus spreader.(P18)

The failure of older people to follow the new social norm of wearing a face mask was correlated with their economic strain:

I also want [to wear] face masks, but they are very expensive now. I have retired and have no income, and I just rely on “fruit money” [Old Age Allowance] and my savings to maintain my livelihood. How can I have spare money to buy face masks? If I have money, I would rather buy food first. If you have no face mask, you will not die, but if you have no food, you will die in 2 or 3 days. I am not afraid of pneumonia [COVID-19], but I am afraid of having no money and no food. I know that I have to wear a face mask when I am not at home, so I usually reuse a face mask if I need to go out.(P20)

##### Not Receiving COVID-19 Vaccines

With the introduction of COVID-19 vaccines, the Hong Kong government and medical authorities were encouraging the public to receive vaccines. Getting vaccinated was advocated as a social responsibility, which has placed pressure on people to receive vaccines:

The government and doctors have been saying that you should receive vaccines because you are protecting not only yourselves but also others. They have been saying that receiving vaccines is a social responsibility, and this is like a form of social pressure. If you do it, then you are a good person; otherwise, you will be labeled as having no civic responsibility.(P22)

Not receiving vaccines became a tangible marker of “deviant others” and “unsafe” people:

I would not intentionally avoid those who have not been vaccinated, but I would consider those who have been vaccinated as safer and so would be more willing to gather or eat with them. I would feel more hesitant to gather with those who have not been vaccinated because they are at a higher risk [of infection with COVID-19]. If they were to be infected, I would be quarantined even though I have been vaccinated.(P64)

Thus, vaccination was encouraged through the construction of “us” (those who were vaccinated and are “safe”) and “others” (those who have not been vaccinated and are “unsafe”) in the community. A participant indicated that some of her friends would only gather with those who had been vaccinated; thus, unvaccinated individuals became secluded and marginalized:

I have some friends who eat with only those who have been vaccinated—they really ask if you have been vaccinated before eating with you. A friend of mine who has been vaccinated really asked me whether I have been vaccinated before eating with me. Because I had not yet been vaccinated, she suggested eating later… Vaccination is no longer a choice for me, but it has become a pressure for me if I still want to eat with my friends.(P41)

The “vaccine bubble” policy that was introduced by the Hong Kong government in late April 2021 and the Vaccine Pass in late February 2022 have further contributed to the marginalization of those who have not been vaccinated against COVID-19:

I have been vaccinated, and now, I can eat with more of my friends at a table according to the vaccine bubble. If we are all vaccinated, then there will be less restriction in eating because we can sit together at the same table… When we organize meal gatherings, we usually only ask those who have been vaccinated to join us. It would be very difficult to eat with those who have not yet been vaccinated because doing so would mean that we are unable to sit together and have to sit at separate tables.(P49)

I can only eat with those who have already finished the vaccine course according to the Vaccine Pass. I have some very good friends and I want to see them, but unfortunately they are not yet vaccinated, so they cannot eat in any restaurants. You know, it is very important to eat together so we can have chit-chat. But because they are not yet vaccinated, they cannot eat in any restaurants, so I haven’t met them for a long time already.(P68)

#### 3.1.3. Social Construction of “Geographical Others”

During the two outbreaks in Hong Kong, “geographical others” were constructed primarily on the basis of the high morbidity rate of certain districts. Living in those regions with a high morbidity rate was perceived as deviant:

I believe that Tsz Wan Shan and Tuen Mun are very dangerous because there have been serious cluster outbreaks in these two areas. In Tsz Wan Shan, many people have been infected at restaurants in shopping malls. If I know that someone is living in Tsz Wan Shan or Tuen Mun, I will avoid them. […] I remember that there were certain dangerous areas that I would avoid during [the] SARS [outbreak] as well, such as Shatin and Amoy Gardens, because there were severe outbreaks there.(P14)

The participants often perceived “geographical others” to be of low socioeconomic status. People who lived in areas that were stereotyped as housing individuals or families of low socioeconomic status had a relatively high tendency of being perceived as “geographical others”:

Rich districts would be safer [than would poor districts] because the people [residing] there would have higher quality and be cleaner. I would avoid going to poor districts. The most dangerous areas are those with many poor people because they would be dirty. The education level [of the people residing in poor areas] would be low, so they do not have much knowledge regarding hygiene. Also, they [poor people] can only work in low-status jobs, such as cleaning and catering, which are associated with high risks of [SARS and COVID-19] infection. Many outbreaks have occurred in public housing estates and old districts because poor people reside there.(P1)

The Hong Kong government’s pandemic control policy possibly reinforced the correlation between “geographical others” and low socioeconomic status. In February 2021, the government introduced a “stay in the original premises” inspection order, enclosing certain communities from 7:00 p.m. to 7:00 a.m. on the following day to launch a “mandatory inspection” for the residents. The enclosure order was mostly implemented in districts with a high number of residents of low socioeconomic status. This order had a stigmatizing effect and resulted in the labeling of “geographical others” as those with a high risk of COVID-19 infection:

All my relatives and friends are scared of me because I live in a building that was enclosed by the government. They knew about this from the news, so I could not hide this [information]. My boss even asked me not to go back to the office for two weeks. Although I tested negative [for COVID-19 during the compulsory testing] and I can go back to work now, my colleagues still avoid me. My relatives avoid me as well, so I could not celebrate Chinese New Year with them this year… I feel that the situation is a bit unfair because enclosure orders are rarely implemented in rich districts, even though infections are found in the rich districts as well. I feel that we are targeted because we are poor. Do you think that the government would dare to enclose the Mid-Levels? I do not think so because the government is afraid of the rich.(P32)

The stigmatization of “geographical others” during the two outbreaks was determined to be related to the capitalist ideology of Hong Kong. “Geographical others” were perceived to be of low socioeconomic status, and they were highly vulnerable to stigmatization. The social construction of “geographical others” not only served as an informal social control mechanism that facilitated pandemic control but also reinforced the existing value system of Hong Kong.

#### 3.1.4. Social Construction of “Occupational Others”

“Occupational others” were considered to be a high-risk group for infection because of the nature of their occupations. The nature of occupations, however, was interlocked with the perceived lower socioeconomic status of those occupations to a large extent:

It sounds like many cluster outbreaks are related to restaurants. Catering workers have to come into contact with many customers every day, and they have to touch things that have been used by different customers. Their hands can easily carry the [COVID-19] virus [and spread it to them]. Also, most of the catering workers belong to a low social class, so they may not be very hygienic and can be potential virus spreaders. Therefore, I do not go out to eat now.(P15)

Another participant also added:

Street cleaners are at a very high risk because they have to touch a lot of dirty things. Although I know that they have contributed a lot to our society, I would still avoid them. Because of their job and because they belong to a low social class, they would find it very difficult to prevent themselves from being infected.(P28)

“Occupational others” also includes healthcare providers because of the nature of their jobs:

Because I am working in a hospital, I had moved to a hotel to avoid bringing the virus home. However, finding a hotel was a very frustrating process. Many hotels would ask you to fill in a health declaration form, asking if you have any symptoms and if you have contacted any infected patients. Because of my job, I would encounter infected patients every day. But if I were honest with them [hotel staff], I would not be able to get a room. It is frustrating. Although many people would say that they support you, in reality, we were not accepted by society during COVID [the COVID-19 pandemic] and SARS. When epidemics come, we [health-care providers] will be isolated.(P30)

However, healthcare providers encountered less and shorter stigmatization than did other “occupational others” during the two outbreaks because they fulfilled the existing social and cultural values of the capitalist ideology:

Health-care workers are dangerous because they are working in hospitals. I am living close to the Prince of Wales Hospital, and you know, this hospital had a severe outbreak of SARS before. This pneumonia [COVID-19] really makes me think about those horrible days of [the] SARS [outbreak]. However, I welcome them [health-care workers] to live here because they are professionals and so can increase the status and quality of this housing area. I trust that they would be very clean and would not spread viruses and bacteria among us [the residents]. After all, they are highly educated and are of high quality, so they should not be too dangerous.(P56)

#### 3.1.5. Social Construction of “Ethnic Others”

Some ethnic groups were considered to be at a high risk of infection and were stigmatized during the two outbreaks. One of the stigmatized ethnic groups consisted of those from developing countries. For example, Filipino and Indonesian domestic helpers were stigmatized as high-risk groups during the two outbreaks:

I believe that Filipino and Indonesian maids are at a very high risk because their home countries are not developed, so their hygiene standards are not high. It is not news at all that the food prepared by these maids can be contaminated. Also, they often get together on Sundays, without any face masks and social distancing. They often leave places in a mess after their gathering, too. The pneumonia [COVID-19] outbreaks in the hostels of these maids further prove that they are at high risk. I have been aware of their risk since [the] SARS [outbreak], so I do not hire any maids since then.(P33)

The stigma attached to Filipino domestic helpers was reinforced after their infection with the mutated N501Y strain of COVID-19 in early May 2021:

It is the third case of mutated virus infections among Filipino maids already. They are dangerous, but it is not a surprise. The Philippines is not a well-developed country, so [Filipino] people’s hygiene standards are not high. The [COVID-19] outbreak there was also very serious. Therefore, you can expect that there would be many virus carriers in this group. You know, they often hang out with other Filipino maids on Sundays, sitting very close to each other without any face masks or social distancing. Whenever I see them these days, I walk away from them. They really have very low awareness, and they still complain about the compulsory virus testing order on them. Because they have low awareness, they do not remain alert regarding infection—this is the most dangerous thing.(P52)

South Asians, such as Indians and Pakistanis, were also perceived as a high-risk group:

India and Pakistan have many cases [of COVID-19]. It is not surprising because India and Pakistan are under-developed. The hygiene standards of these countries and their people are low as well. I dare not go to those places where there are many Indians and Pakistanis, such as Chungking Mansions. Actually, Chungking Mansions have been infamous and dangerous for a long time. This outbreak period is even more dangerous because of these people. They are the poor in Hong Kong and can only work in low-status jobs, which makes them more risk-prone [to COVID-19 infection].(P21)

The stigmatization of South Asians, as well as Filipino and Indonesian domestic helpers, was related to the stereotypes held by the Hong Kong people toward these ethnic minorities, which are often stigmatized as being “not advanced,” “under-developed,” and having “low hygiene,” as popularly indicated by the participants’ responses. In addition, the aforementioned ethnic minorities mostly belong to a low socioeconomic class and work in jobs with low social status; thus, they are vulnerable to stigmatization because they fail to fulfill the cultural value of the capitalist ideology of Hong Kong. They can also be considered “occupational others” and thus experience double stigmatization.

Those who visited Hong Kong from overseas were also considered “ethnic others” during the COVID-19 pandemic because they were believed to be potential virus carriers owing to not only the high infection rate in many countries but also the Hong Kong government’s quarantine order and electronic tracking wristband policy for those who were coming from overseas:

Foreigners are dangerous because pneumonia [COVID-19] is so serious in many Western countries. What I worry the most about are those who are studying overseas and now coming back to Hong Kong for summer vacation. Although they should stay in hotels for quarantine, the news reported that many of them had escaped from quarantine. Therefore, I would pay high attention to see if anyone around me is wearing an electronic wristband. If yes, I would immediately walk away from them. Even many shops refuse to allow people with electronic wristbands to go in. […] Also, many foreigners do not wear face masks properly. They only wear a cloth face mask, or they do not cover their noses, and this causes them to spread the virus. Whenever I see any foreigners who are not wearing a surgical face mask properly, I walk away from them immediately.(P34)

Foreigners became “ethnic others” because they failed to fulfill the new social norms of Hong Kong during the COVID-19 pandemic. However, the stigma attached to foreigners from Western countries was temporary because their cultural violation was temporary and they could fulfill the capitalist ideology. P34 mentioned the following:

I believe that Westerners and students studying abroad are not as dangerous as Indians, Pakistanis, and Filipino and Indonesian maids. They [Westerners and students studying abroad] come from more advanced places [than India, Pakistan, the Philippines, and Indonesia], which are richer and more highly educated, so they should be cleaner. It is just that Westerners do not wear a face mask; otherwise, they are not that dangerous. They are unlike those who are coming from under-developed places.(P34)

### 3.2. Traditional Cultural Values

The traditional cultural values also constituted a prominent informal social control mechanism for certain groups during the SARS and COVID-19 outbreaks in Hong Kong. These values were emphasized and reinforced among the people in Hong Kong, which was favorable to pandemic control.

#### 3.2.1. Glorification and Moral Condemnation of Healthcare Providers

The traditional cultural values of Hong Kong served as an informal social control mechanism for healthcare providers during the two outbreaks. This social control mechanism involved glorification. Healthcare providers were often glorified as “selfless” and “having a sense of sacrifice and courage” during the two outbreaks. Moreover, the Chinese proverb “a healing person [medical doctor] has a parental heart” has long been a cultural expectation for healthcare providers in Hong Kong. The aforementioned social expectations were reinforced and normalized through glorification, which served as a moral control mechanism for them. Under this cultural environment, any complaints and concerns by healthcare providers regarding their personal needs, emotions, and anxieties could be morally condemned:

The doctors and nurses in Hong Kong have a very high quality. We are so fortunate to have these very good doctors and nurses who have sacrificed themselves during the two outbreaks. They are unlike those in Taiwan, where many doctors and nurses were so unprofessional that they escaped from hospitals during [the] SARS [outbreak]. “Doctors have a parental heart,” and leaving patients behind because of the fear of being infected is so unprofessional, selfish, and irresponsible. If you are really afraid of being infected, then you should not work in health care.(P12)

Glorification functioned as an informal social control mechanism regarding healthcare providers; however, the effect was weaker during the COVID-19 pandemic than during the SARS outbreak. Some participants were affected by the capitalist ideology and believed that healthcare providers should perform their healthcare roles as expected because they are well-paid for their jobs:

Without a doubt, I appreciate the work of doctors and nurses in this time of pneumonia [COVID-19]. However, frankly, it is their responsibility, and they are obligated. They are well paid for this, and no one takes advantage of them. It is because of this potential risk that they are well paid as a reward, and this is the core value of a commercial society. They should have understood this risk well when they chose to work in health care. If they are afraid, they can always quit, and no one will force them to continue. If they choose to stay, then they should not complain at all. I agree that they should be respected, but I do not think that we need to overemphasize their devotion.(P5)

During the fifth wave of COVID-19 infections in early 2022, there were considerable reports on private hospitals and private practice doctors about their refusal of treating patients with COVID-19 and those who were just merely having respiratory symptoms [28,29,30]. Much moral condemnation was targeted at private hospitals and private practice doctors in mass media, which served as social control for them and pressured them to collaborate with public hospitals in managing patients with COVID-19. This participant expressed a popular viewpoint:

It is totally unacceptable for these private hospitals and private doctors to refuse pneumonia [COVID-19] patients. They are well paid, and they should have already thought about this risk when they first chose to work in health care. Refusing a patient is immoral, and private hospitals and private doctors should help the public hospitals by treating pneumonia patients as well in this difficult time. “A healing person should have a parental heart”. I am so frustrated that our doctors and nurses nowadays have become so different from those in the SARS outbreak.(P55)

At the beginning of the COVID-19 pandemic, a group of healthcare providers from The Hospital Authority Employees Alliance organized a strike against the government’s cross-border pandemic control policy. However, due to condemnation from the public, this strike was soon called off. The condemnation from the public, which was based on the existing cultural expectations for healthcare providers in Hong Kong, served as an informal social control mechanism for manipulating the behavior of healthcare providers during the COVID-19 pandemic:

I appreciate the doctors and nurses who worked during [the] SARS [outbreak] much more than those who worked in this pneumonia [COVID-19] outbreak. During [the] SARS [outbreak], none of the doctors and nurses ever complained; they just kept working very hard. However, during the current pneumonia outbreak, some doctors and nurses have not been devoted. They have many complaints and even went on strike. I believe that, if the public had not condemned their strike, it [their strike] would have still be on now. As doctors and nurses, they should consider patients their top priority because “a healing person should have a parental heart.” If they are dissatisfied, they can just quit, but they should never make use of patients to achieve their goals.(P66)

#### 3.2.2. Patriarchal Values against Homemakers

The traditional gender value of women as homemakers, which is embedded in Chinese patriarchal belief, was reinforced during the SARS and COVID-19 outbreaks. This cultural expectation for women served as an informal social control mechanism for them, motivating them to fulfill domestic expectations such as engaging in frequent household disinfection, thus facilitating pandemic control measures. Most of the female participants in this study had internalized the aforementioned gender expectation because they believed that they had to bear the full responsibility of taking care of their family members and maintaining hygienic conditions in their homes:

I feel that the [periods of the] SARS and pneumonia [COVID-19] outbreaks were the toughest time for me. As a housewife, I have no excuse, and I have to work harder to keep the home clean. No one thinks about how terrible the life of a housewife can be during pandemics. I have to do more cleaning and disinfection at home. If any of my family members were to get infected, I would be responsible, and others may think that I did not do my job well. The situation is worse under the pneumonia outbreak because the pressure is much heavier than that under [the] SARS [outbreak]. During the SARS [outbreak], I only had to do more home cleaning. However, during [the] pneumonia [outbreak], I have had to not only do more home cleaning but also take care of my husband and children, all of whom are working or studying from home. The situation is similar to “closed-cage beast fighting” [a Cantonese slang term meaning that several people are struggling together within a small area]. I did not ask for my husband’s help because he would expect it [household work] to be my duty. I believe that all people think it is normal for a housewife to do all these [tasks]. After all, as a housewife, this is what I am expected to do.(P25)

Working women were suffering from the double burden of their domestic role and external work:

I have been much busier during [the] pneumonia [COVID-19] [pandemic] than [I was] during [the] SARS [outbreak]. During [the] SARS [outbreak], I only had to do more cleaning at home, and I could still go out to work to have some hours off from family duties. However, during [the] pneumonia [pandemic], there has been far more for me to do because I have to work from home. In addition to office work, I have to do cleaning and disinfection at home and take care of my husband and two children simultaneously, so I have to perform at least three tasks simultaneously. I am the only one to keep my children busy until they have gone to sleep at night, but then I have to do my office work and housework. I am so tired because I have been working for almost 24 h a day throughout the pneumonia [pandemic]. No one helps me, not even my husband, because he believes that those [household tasks] are my duties.(P23)

The traditional gender value of women performing domestic roles was reinforced during both outbreaks. The outbreaks not only reinforced the existing social and cultural expectations for women in Hong Kong but also justified and normalized the burden (or double burden) on them because of their internalization of these expectations.

## 4. Discussion

In contrast to the previous literature showing that social and cultural values often serve as obstacles to successful epidemic control [12,13,14,17,18], this article demonstrates the opposite in which social and cultural values also serve to favor epidemic control. Several studies have indicated that the fear of contracting an infection as well as the perceived risk, perceived susceptibility, and perceived benefits have major influences on people’s practice of preventive health behaviors [31,32,33,34]. However, the findings of the present study provide a different and more macro perspective. The stigmatization and discrimination against “others” serve as informal but powerful social control mechanisms for motivating people’s practice of preventive health behaviors during the SARS and COVID-19 outbreaks. Traditional cultural values on certain social groups such as healthcare providers and homemakers were also reinforced, justified, and normalized in favor of epidemic control.

The stigmatization and discrimination of “others” as well as the traditional cultural expectations on certain social groups during the SARS and COVID-19 pandemics were rooted in the capitalist ideology that has long been adopted by society in Hong Kong since colonial times. The structural transformation of the Hong Kong economy to export-oriented industrialization in the 1950s led to significant changes in Hong Kong’s occupation and class structures, leading to the rise of the middle class [35]. The middle class in Hong Kong symbolizes the ability to achieve upward mobility through higher education and self-efforts, is able to enjoy economic affluence, and has become a distinctive social group with a sociocultural identity [35]. Although Lui (2003) argued that the development of the sociocultural distinction of the middle class from low social classes is not well established in Hong Kong, the sociocultural distinctions to maintain distance from the low social classes have become more obvious among the middle class during the two epidemics considered in this study. As demonstrated by the participants, the different groups of “others” constructed and the traditional cultural values emphasized during the SARS and COVID-19 outbreaks were embedded in the capitalist ideology and such ideology was reinforced throughout both epidemics.

The findings of this study, thus, are in agreement with the framework of Critical Medical Anthropology, which suggests health issues should be understood within the context of social class relations inherent in capitalism [36]. Social inequality and power, according to Baer et al. (1997), are inevitable in capitalism and they are the key determinants of most health issues [36]. The social construction and stigmatization of the different groups of “others” during the SARS and COVID-19 pandemics were a demonstration of social inequality, in which those who are of low socioeconomic status and/or those who are the less powerful are subject to the control, and perhaps exploitation, of those coming from high socioeconomic hierarchy and/or those who are the more powerful in Hong Kong.

Past experiences of the SARS outbreak remarkably shaped the participants’ perceptions and behaviors toward the COVID-19 pandemic. The behavioral responses of the participants during the COVID-19 pandemic—which involved the social construction and stigmatization of different groups of “others”—were embedded in their experiences during the SARS outbreak as well as the traditional cultural values of Hong Kong. “Others” are social deviants who do not behave according to social and cultural values but have not necessarily done something wrong [16]. People are socially constructed as “others” when their attributes “cross the lines of social structures” [37], which causes them to fail in fulfilling existing social and cultural norms. In line with the finding of Hays (2003) [14] and Farmer [13,15], the participant’s responses toward a new infectious disease often involved the stigmatization of marginal social members as “others.” The social construction of “others” served as an informal social control mechanism to manipulate people to engage in favorable health behaviors for pandemic control. Moreover, the social construction of “others” was based on the embedded social and cultural values of Hong Kong, which were reinforced during the two outbreaks considered in this study. According to the participant responses, “diseased others,” “deviant others,” “geographical others,” “occupational others,” and “ethnic others” failed to fulfill the social and cultural norms of Hong Kong before and/or after the outbreaks began. In many cases, these norms were in line with the embedded ideology of capitalism in Hong Kong. The failure of people to fulfill the embedded ideology of capitalism in Hong Kong made them vulnerable to being scapegoated and stigmatized during both outbreaks.

The results of this study are in agreement with those of previous studies showing that people infected with SARS or COVID-19 were often stigmatized [38,39,40]. The social construction of “diseased others” was a powerful social control mechanism for motivating the participants to conform to the control measures adopted during the two outbreaks. Infected individuals were stigmatized not only because of their infection but also because they were considered to possess negative attributes that violated the existing social norms of Hong Kong, such as low civic responsibility, low social consideration, and low socioeconomic status. Moreover, the collective responsibility for COVID-19 created by the quarantine policy contributed to the social construction of “diseased others,” which resulted in participants experiencing social and moral pressure to not involve others in problems caused by the outbreaks. This phenomenon could increase people’s motivation to conform to pandemic control measures. In addition, those who were not necessarily getting infected but exhibiting respiratory symptoms were secluded from some medical systems. Such difficulty in getting treatment during the COVID-19 outbreak was also an informal social control that could encourage people to avoid getting sick and becoming “diseased others,” which favors epidemic control.

Although the stigmatization of “diseased others” could motivate people’s adoption of preventive behaviors that favor epidemic control, this creates equity problems for those social groups who are more vulnerable to becoming “diseased others.” Consistent with the results of previous research [41], the present study findings indicate that participants belonging to a low socioeconomic class found it difficult to follow the preventive health measures suggested by the government and medical authorities in Hong Kong. Materially speaking, the participants who were coming from a low socioeconomic class found difficulty in purchasing infection control materials due to their economic strain. Socially speaking, they found difficulty in fulfilling the infection control measures, as they possessed little bargaining power in their jobs and few social resources; hence, the advocacy of working from home, staying at home, and eating takeout were all considered luxuries. Because of the social inequality experienced by the aforementioned participants, they had to experience health inequality that caused them to suffer from a high risk of infection, which increased their vulnerability to stigmatization.

“Deviant others” were socially constructed primarily on the basis of the new social norms created during the two outbreaks and comprised older adults, those who did not use a face mask in public areas, those who did not follow the stay-at-home and takeout policies, or those who were not vaccinated against COVID-19. These new social norms were unfavorable to people of low socioeconomic status. Using face masks is an additional financial burden on the aforementioned people, and staying at home may be infeasible for them because their jobs and living environment mostly do not support working from home. Dining in was also the only option for these people because their living and work environments made it difficult for them to eat takeout. Moreover, the older adult participants failed to fulfill capitalist values because of their retirement, which led to a decline in their socioeconomic status and an increase in their social dependence. In summary, people of low socioeconomic status were vulnerable to stigmatization as “deviant others” during the two outbreaks because their economic situation prevented them from conforming to the new social norms and embedded capitalist ideology of Hong Kong.

The new social norms developed during the outbreaks, such as wearing a face mask, following the stay-at-home and takeout policies, and receiving COVID-19 vaccines, were associated with social responsibility, and failure to fulfill these norms was perceived as “deviancy”. The Hong Kong government’s “vaccine bubble” and the later Vaccine Pass policies during the COVID-19 pandemic resulted in unvaccinated people being stereotyped as “risky” and “dangerous,” and unvaccinated people were more vulnerable to experiencing social seclusion than vaccinated people. The two outbreaks also served to remind people of perceived negative attributes in society. For example, smoking was reinforced as a deviant attribute during the COVID-19 pandemic because smoking violated the new social norm of wearing a face mask as well as because of the embedded appeal of antismoking at an institutional level. The stigmatization and social seclusion of “deviant others” became an informal social control mechanism, which resulted in considering the adoption of pandemic control measures as a preferable social attribute.

In line with the findings of previous studies [42,43], the findings of the present study revealed that those residing in high-morbidity regions, those working in certain occupations, and those belonging to ethnic minority groups were highly vulnerable to stigmatization during the SARS and COVID-19 outbreaks. The social construction of “geographical others,” “occupational others,” and “ethnic others” was based on the embedded social values of capitalism in Hong Kong, which creates equity problems for these disadvantaged social groups. The participant responses indicated that districts with a large population with low socioeconomic status were perceived as being “risky” and were thus highly vulnerable to stigmatization. Those occupations perceived to be of low socioeconomic status were highly vulnerable to long-lasting stigmatization as “dangerous” because they failed to fulfill the embedded capitalist ideology of Hong Kong. Ethnic minorities from developing countries such as the Philippines, Indonesia, India, and Pakistan were stereotyped as “high-risk” groups because they are poor and not well-educated and mostly work in jobs associated with a low social status. Although studies have found that healthcare providers are stigmatized and discriminated against during outbreaks [39,44,45,46,47] and show consistency with the findings of this study, the present study found that healthcare providers encountered temporary and relatively mild stigmatization. Furthermore, people from relatively rich Western countries encountered considerably less stigmatization than did other vulnerable groups, which is attributed to those countries’ successful fulfillment of the capitalist ideology. Although the construction of “geographical others,” “occupational others,” and “ethnic others” favor epidemic control in the sense that people could become more alert by avoiding the risky groups, this also creates an equity issue, as these socially disadvantaged groups are more subject to stigmatization.

Finally, although the traditional cultural values expected of healthcare providers and homemakers favored epidemic control in the two outbreaks, these expectations had become a latent social control and functioned as moral and behavioral guidance for these social groups. When these expectations had become normalized and justified, these social groups could be exploited. As demonstrated by the participants who were homemakers and from the participants’ expectations of healthcare providers, their internalization of these traditional social and cultural values can become a moral control for them, inhibiting them from voicing their concerns. This may thus result in an equity issue for these social groups.

## 5. Limitations

Although all participants experienced the two outbreaks, memory errors might have been associated with their experiences during the SARS outbreak, which occurred approximately two decades ago.

## 6. Conclusions

In contrast to previous studies that found that social and cultural values can be obstacles to epidemic control, the present study revealed that the social construction of five groups of “others” during the SARS and COVID-19 outbreaks, the reinforcement of the traditional cultural values of Hong Kong during these outbreaks, and the existing social and cultural values based on the capitalist ideology of Hong Kong, facilitated pandemic control at the community level. These two outbreaks reinforced the embedded social and cultural values of the capitalist ideology of Hong Kong, which increased the vulnerability of disadvantaged social groups to stigmatization. Individuals of low socioeconomic status found it difficult to conform to pandemic control measures, which resulted in a vicious cycle of stereotypical correlation between low socioeconomic status, deviant behaviors, and high infection risk during the two outbreaks.

## Data Availability

The data presented in this study are available from the corresponding author on request. The data are not publicly available due to the confidentiality of the participants.

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
