# Peer review of "The Role of Social and Cultural Values in Pandemic Control in a Chinese Community: An Ethnographic Study on the Construction and Stigmatization of “Others” in Severe Acute Respiratory Syndrome (SARS) and COVID-19 in Hong Kong"

_ijerph, 2022, doi:10.3390/ijerph192013517_

Round 1

Reviewer 2 Report

It would be very interesting to use tables and figures to illustrate everything in a better way.

Reviewer 3 Report

This is an extended and in depth qualitative exploration of  social and cultural values that can be stimulated into positive preventive strategies at a time of a health threat. It is based on an exploration of the response in Hong Kong to the SARS and later COVID19 outbreaks. It is a long and detailed  paper on an important and topical issue. It should be of great interest to  the journal readers.

The paper  draws on major studies undertaken by the author during the earlier SARS epidemic and the more recent COVID epidemic. The core issues examined are drawn from qualitative analyses using participant observation and interviews. Themes are centred around the issue of the “other “ and “the construction and stigmatisation  of others that derives from increased control by informal social control mechanisms”.  These sections of the paper are well developed and make a major and timely contribution to knowledge in this new and concerning field of public health.  My reading is that the paper also builds an argument that the capitalist ideology of Hong Kong is/was closely linked to  the strategies used to manage the epidemic and that this lead to serious stigmatization of disadvantaged and ethnic groups. Given the findings that are developed and related to socioeconomic groups I wonder if there are figures that reflect this in the associated public health COVID19  statistics

This is an important and controversial finding and there is a need for definition and references to develop the reader’s understanding of  the characteristics of “capitalist ideology” as it is constructed in Hong Kong.

The ethnographic methodology used is well described and is applied in such a way as to extend and inform the reader in a new and valid understanding.   The large section on the author’s personal “reflexivity” in the research and the management of Ethics approval  are to be commended.

 I was  concerned about the management of the sampling used for the interviews.  My reading  is that the two samples are very different . The smaller  group (20) is a university sample of students or academics and the larger sample is from labour and trade union community groups (50). I think that  the issues raised could sometimes  reflect these two diverse samples and would have been interested to have seen the present quotation attributions to numbered participants  ( P1 -P69 ) also give an indication as to which sub-sample the respondent belongs.  I am aware that this attribution could be a problem for anonymity and if this is the case believe it would be helpful to mention in the methodology. This perhaps is of particular importance because of the conclusions and insights related to social disadvantage.

 I  found the development and comments related to the “Ethnic other group” to be  confronting but accept that this is an important finding.  However, I wonder if there was any information about the ethnic  group membership (if any) of the participants that could be described in the sampling methodology. The reference to “pneumonia COVID 19” throughout the article is of concern . Perhaps it is the way it is referred to in Hong Kong but if that is the case it should be mentioned specifically in the introductory areas that it  is incorrect but is being used in quotation. The accepted  understanding is that the appropriate term  is COVID19 and while a serious complication of the illness can be pneumonia it is by no means a  normal   or expected associated syndrome.

It is an important paper and should be of considerable interest to readers in the public health field.

---------

The author needs to use the journal’s format advice and indent the sections of the paper that are quotations to distinguish them from the author’s comments.  The failure to do this adds reading  difficulty.

Some sections which I assume draw on the author’s edited  thesis are clear but other paragraphs do need to be proof read by a professional editor …..

Line 543 uses (COVID_G5) as the quotation attribution.

Lines 786-788 on private hospitals is controversial and  needs referencing.
